# Psychometric Analysis and Contribution to the Evaluation of the Exams-Related Emotions Scale in Primary and Secondary School Students

**DOI:** 10.3390/ijerph19116770

**Published:** 2022-06-01

**Authors:** Ana Isabel Obregón-Cuesta, Luis Alberto Mínguez-Mínguez, Benito León-del-Barco, Santiago Mendo-Lázaro, Jessica Fernández-Solana, Josefa González-Santos, Jerónimo J. González-Bernal

**Affiliations:** 1Department of Mathematics and Computing, University of Burgos, 09001 Burgos, Spain; aiobregon@ubu.es; 2Department of Education Sciences, University of Burgos, 09001 Burgos, Spain; 3Department of Psychology and Anthropology, University of Extremadura, 10071 Caceres, Spain; bleon@unex.es (B.L.-d.-B.); smendo@unex.es (S.M.-L.); 4Department of Health Sciences, University of Burgos, 09001 Burgos, Spain; jfsolana@ubu.es (J.F.-S.); mjgonzalez@ubu.es (J.G.-S.); jejavier@ubu.es (J.J.G.-B.)

**Keywords:** emotions, exams, educational context, students, primary school, secondary school, gender, academic performance

## Abstract

The objective of this research was to perform a construct validity, and a psychometric analysis of the Exams-related Emotions Scale (EES), analyze the differences in their scores regarding gender and academic year in primary and secondary education student, as well as study the relationship between emotions and academic performance. During the construction and analysis of the psychometric characteristics of the scale, an Exploratory Factor Analysis (EFA) and a Confirmatory Factor Analysis (CFA) were performed. To test EES scores based on gender and school year, the T-Student test for independent samples and ANOVA were used, respectively. To verify the relationship between the scores of the different types of emotions and academic performance, the Pearson correlation test was performed. A total of 562 students belonging to the 5th (*n* = 228) and 6th (*n* = 186) primary school year and the 1st (*n* = 134) and 2nd (*n* = 94) secondary school year participated in the research. Age between 10 and 15 years old (mean = 11.66, standard deviation = 1.206) both males (50.5%) and females (49.5%). The results offered support for the three-factor structure. The analysis of invariance with respect to gender showed that the factor structure was invariant. Boys scored higher on the positive emotions factor and lower scores on anxiety than girls. Regarding school year, negative emotions and anxiety related to exams increase in the secondary courses compared to Primary while for positive emotions it is primary school students who obtain higher scores. The correlation coefficient between negative emotions (as well as anxiety) and school performance is negative whereas the correlation coefficient between positive emotions and academic performance is positive. EES scale is an instrument with scientific rigor and with adequate reliability and validity to be able to know the emotions that primary and secondary school students suffer when they are subjected to evaluation processes in the academic context.

## 1. Introduction

Various investigations contemplate the situation in which students find themselves when they are subjected to an evaluation test or an exam. Facing a situation like this is a complex process in which various cognitive and behavioral strategies are involved with the sole purpose of managing the demands, both internal and external, that are placed an individual [1,2,3].

Most students, when they are at exam time, tend to experience high anxiety, which can also have a negative impact on academic performance and their health [4,5,6,7]. In Spain it is estimated that between 15 and 25% of primary and secondary school students have high levels of anxiety before an exam. Confirmed, likewise, in the study of Hernández [8] where the presence of anxiety before an exam, and possible need for specialized help, in 20.84% of a sample of 28,559 students from 16 Spanish universities is highlighted [8,9].

Anxiety can be described as an emotional manifestation with a multicausal origin. This can be triggered by the negative perception of an event, where the person focuses on the possible harmful consequences. Likewise, it can have negative repercussions on the individual, cause a probable academic failure, decrease the self-confidence and motivation of the student, generate a fear of criticism, and in some cases even the abandonment of the subject or complete academic year; being able to interfere with cognitive processes and interpersonal relationships [2,9,10,11,12,13,14]. Anxiety about an exam is manifested through a series of negative emotional reactions that constitute a serious problem, due to the great impact it has on performance, as well as the high percentage of students who suffer from it. It is very important to detect school failure due to this circumstance, and not to problems related to learning [4,9,15].

The relationship between anxiety and student performance has been a major research objective in recent years, but minimal attention has been paid to other emotions. Throughout the twentieth century, studies have been carried out in which emotions are related to exams, and to academic failure or success, although these are very few and far between [16,17]. The emotions felt by students are of great importance for the short- and long-term development of learning processes, the genesis of interest and the acquisition of knowledge. The emotions suffered by students are multiple and very varied, with a lack of research in all those, both positive and negative, different from anxiety [16,17].

According to the Three-Dimensional Theory of Anxiety, the emotional reactions that occur can be observed on three levels; subjective cognitive (experience), physiological (bodily changes) and motor (observable behaviors). These manifest, for example, in feelings of discomfort, worry, negative thoughts, increased heart and respiratory rate, muscle tension, hyperactivity or crying, among others [9,18].

It is known that in the face of both a negative and positive emotion, a series of strategies are generated at the cognitive level to regulate them. These strategies of a cognitive nature are called cognitive emotion regulation strategies (CERS). CERS can help modulate an emotional response to a stressful situation through a cognitive approach to emotional information received by the person [2,19,20]. 

Emotional regulation can be defined as the intrinsic and extrinsic processes that are responsible for evaluating and modifying emotional reactions, whether positive or negative, with the aim of fulfilling a personal goal [2,20,21,22,23]; and that is undoubtedly related to emotional intelligence [2,19]. Emotional intelligence has been described as an important factor in achieving academic success. It is directly linked to the control of emotions, the management of frustration, academic performance, and emotional flexibility among others. Therefore, it is concluded that the deficit in emotional intelligence skills can have a special impact on students [19,24,25].

To detect existing deficits in the students’ emotional regulation when undergoing an exam, reliable and valid assessment instruments aimed at specific groups are required. Considering the great relevance of this aspect in the educational context, the main objective proposed is to perform a construct validity, and a psychometric analysis of the Exams-related Emotions Scale (EES) in a sample of primary and secondary school students. Thus, we propose the following hypotheses: (1) the EES is a tool with scientific rigor and adequate validity and reliability to assess the emotions experienced by students in primary and secondary education when taking exams; (2) there are differences in the emotions experienced around the exams according to gender (with boys experimenting higher positive emotions and lower anxiety than girls) and school year (increasing negative emotions an anxiety with years); (3) Negative emotions and anxiety negatively affect academic performance, while positive emotions positively affect it.

## 2. Materials and Methods

### 2.1. Participants

The sample consisted of 562 students of Compulsory Primary Education (EPO) and Compulsory Secondary Education (ESO), aged between 10 and 15 years (Mean = 11.66; Standard Deviation = 1.21). Distributed in 284 subjects belonged to the male gender (50.5%) and 278 to the female (49.5%). EPO students (*n* = 334) were fifth (*n* = 228) and sixth (*n* = 186), and ESO students (*n* = 148) were in the first (*n* = 134) and second year (*n* = 94). The sample was collected from 5 different centers, public (*n* = 4) and private (*n* = 1), in the Autonomous Community of Castilla y León.

The selection of the sample was made by conglomerates.

### 2.2. Instruments

#### Exams-Related Emotions Scale (EES)

This scale was designed with the aim of evaluating the emotions that students experience when taking exams. The scale consists of 31 items in Likert format with five intervals in numerical form from 1 (Never) to 5 (Always). The items were intended to adequately show the most relevant contents of the construct to be evaluated (Appendix A).

In this sense, the scale evaluates both positive and negative emotions associated with the different moments (before, during and after) of the performance of a test or exam [17,26]. Emotions such as hope, enjoyment, pride, anger, anxiety, shame, or hopelessness. Traditional assessment has considered only anxiety as an emotion in test situations. With the EES scale the evaluation of emotions is extended by also taking into account motivational components and analyzing other emotions experienced by students, both positive and negative, in environments of academic achievement [16].

### 2.3. Procedure

Following the ethical guidance of the American Psychological Association regarding consent, confidentiality and anonymity in the responses, the directors of the educational centers were first contacted and explained the objectives of the research.

Once the collaboration was accepted, the participants were contacted in the classrooms, and after obtaining the informed consent, they proceeded to fill in the scales. Its completion was anonymous, guaranteeing the confidentiality of the data obtained and its exclusive use for research purposes. The administration of the scales was carried out during school hours, offering the pertinent instructions. The anonymous nature of the investigation was emphasized. The questionnaires were completed individually in a suitable environment and without distractions. The process of completing the questionnaires lasted approximately 15 min.

No data was lost as the data was collected using an online form in which all responses are mandatory.

No questionnaire was rejected.

All the students of the grades selected by the principals were included in the research. No boy or girl was excluded based on their culture, language, religion, race, disability, sexual orientation, ethnicity, gender or age.

The Bioethics Committee of the University of Burgos approved the research, (Reference UBU 032/2021), respecting all the requirements established in the Declaration of Helsinki of 1975.

### 2.4. Data Analysis

Initially, for the construction and analysis of the psychometric characteristics of the scales, an exploratory factor analysis (EFA) was performed. Once the EFA was performed, the factorial structure found was confirmed by means of a confirmatory factor analysis (CFA). The reliability of the scale factors was calculated through Cronbach’s alpha, Composite Reliability coefficients, McDonald’s Omega and Extracted Mean Variance. The EFA was carried out using the SPSS-21 program; and for the AFC the AMOS-21 program was used.

To compare the scores obtained in the EES according to gender, the parametric independent samples T test was used, while the analysis according to the academic year was carried out using the one-factor ANOVA test. To study the correlation between the different scores of the emotions obtained with the EES scale and school performance, the Pearson test was performed. A statistical significance value of *p* < 0.05 was established using SPSS version 25 software (IBM-Inc., Chicago, IL, USA).

## 3. Results

The original sample (*n* = 562) was divided into two randomly extracted subsamples (n_1_ = 276 and n_2_ = 276). The first (n_1_) was used to perform the EFA and the second (n_2_) as a validation sample in the CFA. Both subsamples were equivalent as a function of gender, χ^2^(1) = 0.359, *p* = 0.549, and age, *t* (560) = 0.285, *p* = 0.776

### 3.1. Exploratory Factor Analysis of the Exam Emotions Scale (EES)

The sample adequacy measure (KMO = 0.828) and Bartlett’s sphericity test (χ^2^ = 1826.282(66), *p* < 0.001) justified the factor analysis. Using *Kaiser’s rule* [27] eigenvalues greater than the unit and the extraction method of main components with oblimin rotation, a solution of three factors was obtained (Table 1) that together explained 53.6% of the variance.

In the case of oblique rotations, we start from the assumption of correlation between the new factors. *Obllimin* rotation has been used as it allows to establish hierarchical relationships between the factors.

The first factor, “*Negative Emotions”* (12 items) explained 35.4% of the variance and collected information about the negative emotions that are experienced when performing an exam, emotions such as shame, hopelessness, or anger. Some items examples of this factor: “*Before the exam I get depressed because I feel that I do not have much hope of passing the exam”*; *“During the exam I get angry”*; *“After the exam I feel embarrassed.”* The second factor *“Positive Emotions”* (12 items) explained 12.2% of the variance, and collected information about the positive emotions experienced when performing an exam: hope, pride, enjoyment... Some items: *“Before the exam I am so proud of how I prepared that I want to start the exam right now”*; *“During the exam I am happy to be able to face the exam”*; *“After the exam I am brimming with enthusiasm.”* The third factor “*Anxiety”* (7 items) explained 6% of the variance and collected information related to anxiety before the exams. *“When the exam begins, my heart begins to accelerate”*; *“During the exam I am very nervous”*; *“During the exam I am worried about whether I will pass the exam.”*

Cronbach’s alpha of factors 1 (α = 0.915), 2 (α = 0.892) and 3 (α = 0.866) demonstrated good internal consistency.

### 3.2. Confirmatory Factor Analysis

The CFA of the Exams-related Emotions Scale is performed with the second subsample (n_2_ = 272). It is intended to confirm the structure of three factors found in the AFE and if these are related or independent.

Using the maximum likelihood method and taking into account some of the most commonly used adjustment indices (χ^2^, χ^2^/g.l, CFI, TLI, RMSEA y SRMR), four models (M) of emotions related to exams are tested: M1: Three related factors; M2: Two related factors that brings together the first two factors and one independent (Anxiety); Two related factors that bring together factors two and three, and one independent (Positive Emotions); M4 Three independent factors. For optimal adjustment the χ^2^ must acquire non-significant values (*p* > 0.05) [28], el χ^2^/g.l, is considered acceptable when it is less than 5 [29], values higher than 0.90 of the incremental indices (CFI and TLI) and ≥ 0.08 of the RMSEA [30], and the SRMR [31], are considered acceptable.

All models (Table 2) have a significant chi-square value (*p* < 0.05). Models 2 and 4 are discarded, since the values of significant chi square (*p* < 0.05) and the CFI, TL, RMSEA and SRMR indices do not present suitable values. The CFI, TLI indices of models 1 and 3 have values equal to or greater than 0.90, with model 1 being the one that presents a better fit, with a value of χ^2^/g.l, lower and higher indices of adjustment CFI and TLI and values less than 0.07 of the RMSEA and SRMR indicators (Table 2).

The t-values (range 6.55 to 13.09) of the non-standardized regression coefficients of model 1 are statistically significant. The range of standardized coefficients of factor one EP (0.487–0.814), two (0.548–0.813) and three (0.573–0.815), show that the indicators are consistent for the measurement of constructs, these being clearly related (Table 3).

Standardized coefficients of the three factors of emotions before the exams.

### 3.3. Gender-Invariant Analysis

Next, a multigroup analysis is performed to determine if the model of three related factors is gender invariant (146 females and 126 males). The comparison shows no differences *p* < 0.05 in the chi-square value between the different models and the values found in the ΔCFI in the unrestricted model with differences of less than 0.01 of the CFI indices between the four models (Table 4), indicate that the factorial loads of the questionnaire are equivalent for girls and boys.

The coefficients of Average Variance Extracted (AVE) and Composite Reliability (CR) indicate that the model of three related factors presents sufficient evidence of reliability (F1 [4 items]: AVE = 0.519. CR = 0.896; F2 [4 items]: AVE = 0.524, CR = 0.813; F3 [7 items]: AVE = 0.477, CR = 0.863.)

### 3.4. Analysis of the EES by Gender

The results of the inferential analysis showed statistically significant differences in positive emotions and anxiety based on gender, but not in negative emotions (Table 5). More specifically, male participants scored higher on the positive emotions factor (*p* = 0.008) and lower scores on anxiety (*p* < 0.001).

### 3.5. Analysis of the EES According to the School Year

Statistically significant differences were found depending on the academic year in all factors (Table 6). In general, negative emotions related to exams increase in the Secondary courses compared to Primary. As for positive emotions, the opposite occurs, and it is primary school students who obtain higher scores. Finally, regarding anxiety, the same pattern is repeated as with negative emotions, with older Secondary students obtaining higher scores than Primary students.

The data were subjected to the Bonferroni post hoc test to check between which groups the differences detected in the ANOVA test are established. In the negative emotions factor, the main differences are established between 1st year of Secondary with 5th grade of Primary (I-J = 3.68233; *p* = 0.019) and with 6th of Primary (I-J = 5.26007; *p* < 0.001). Also, between 2nd year of Secondary and 6th of Primary (I-J = 5.18211; *p* < 0.001). Regarding the positive emotions factor, significant differences were observed between the means of scores of 1st of Secondary with 5th of Primary (I-J = −4.16226; *p* = 0.001) and with 6th of Primary (I-J = −5.27219; *p* < 0.001). Likewise, among the means of scores of 2nd year of Secondary with 5th of Primary (I-J = −5.59129; *p* < 0.001) and with 6th of Primary (I-J = −6.70121; *p* < 0.001). Finally, in relation to the third factor anxiety, the differences are established between 1st of Secondary and 6th of Primary (I-J = 2.80067; *p* = 0.003).

### 3.6. Analysis of the EES According to the School Performance

Statistically significant correlation is observed (Table 7) between all types of emotions and academic performance. The correlation coefficient between negative emotions and school performance is negative, indicating that, the higher the negative emotions, the lower the performance or average grade (r_560_ = −0.406; *p* < 0.001). The same happens with anxiety, the greater the anxiety, the lower the performance or average grade. (r_560_ = −0.212 *p* < 0.001). In contrast, the correlation coefficient between positive emotions and academic performance is positive, indicating that, the higher the positive emotions, the higher the performance or average grade. (r_560_ = 0.385; *p* < 0.001).

## 4. Discussion

The main objective of this study was to perform a construct validity, and a psychometric analysis of a questionnaire to evaluate the emotions felt by students in primary and secondary education when they are faced with an evaluation situation or exam. The relevance of the study lies in the need to develop an instrument for this specific population in the educational context, since academic performance can be affected by the manifestation of a series of negative emotional reactions to exams, such as the appearance of anxiety [9,10,14,25]; this circumstance may lead to school failure, interference in cognitive processes and interpersonal relationships [4,21,32].

The results of the exploratory and confirmatory factor analysis performed showed a good factor structure, internal consistency, and validity of the instrument. In addition, the EES is gender invariant, so it is an appropriate instrument for the evaluation of exam-related emotions in primary and secondary education students.

To date, there is no specific instrument that evaluates emotions in the face of the examinations suffered by children and adolescents in primary and secondary education in the Spanish school context, with the importance of emotional regulation in these stages of development [33,34].

There are several scales that measure stress, anxiety, or emotional regulation, and that have been used in many cases in the academic context but focused mainly on university-level students. It can be found the Pre-Examination Anxiety and Uncertainty Coping Scale (COPEAU) [5]; Academic Stress Inventory (IEA), validated in Spain and designed specifically for the evaluation of academic stress in university students [9]; Questionnaire of emotional regulation for children and adolescents (ERQ-CA), validated in the Spanish population in the school context between 10 and 19 years [35,36]; Depression Anxiety Stress Scale (DASS-21) [37]; Scale of Cognitive Anxiety against exams (S-CTAS), Spanish version in university students [7,38,39]; Inventory of exam anxiety (TAI), validated in Spanish and adapted to university students [40]; Inventory of anxiety against the exams-state (STAI), version in Spanish with very good validity to measure anxiety before the exams in university students [41,42]; and Westside Test Anxiety Scale [43].

Likewise, it should be noted that the main contribution of this project is that it allows the valid and reliable evaluation of emotions before the exams in the educational context of primary and secondary education, in order to intervene and improve variables such as hope, joy, anxiety, pride, anger, shame or hopelessness when the student faces a situation of academic evaluation [17].

It is important to highlight Pekrun’s Theory of Value of Controlling Emotions [44,45], as it indicates that emotions and values related to achievement can influence students’ learning processes and academic performance [17,46]. In turn, the expression of different emotions is directly related to the academic self-concept of each student; defined academic self-concept as the self-perception of the general capacity in a specific field through personal experience and interpretation of the environment [46].

In previous research conducted with samples of students from both the university level and from secondary and primary education, a statistically significant relationship was found between the level of anxiety and emotions with the situation of facing an exam [9,35,47,48,49]. In addition, a review with meta-analysis allows to clarify that between the ages of 11 and 14, students show a greater negative relationship between anxiety about exams and academic performance compared to upper secondary and university courses [49].

Regarding the second objective raised in the research, based on the analysis of the differences between gender, regarding the development of emotions before the exams, the results show that there are statistically significant differences in terms of positive emotions and anxiety depending on gender, but not with respect to negative emotions. Male students scored higher on positive emotions and lower scores on anxiety. These results coincide with those obtained in the study by Álvarez et al. [9], in which it also finds significant differences in terms of gender, with the female gender having the highest score in anxiety indexes and negative emotions, despite having a higher level of resources to face them. At the same time, in another study where these same results are also obtained, better academic results are evidenced in the female gender compared to the male gender [7]. In the study of Gao et al. [37] Higher levels of depressive symptoms in males are also evident.

In regard to the academic year, statistically significant results were also found, in such a way that negative exam-related emotions increase as the course is higher [50]. In the same way, anxiety increases in secondary courses compared to primary as can be corroborated in recent research [49]. Positive emotions on the other hand are predominant in primary education students with respect to those in secondary education. The change of school year and the transition from primary to secondary school has been a topic of interest for many years for people in the school environment and for researchers, as it represents a change of great importance for the lives of students. It has been observed that school year change is usually well tolerated by most students but that their happiness and well-being clearly decreases when transitioning to secondary school [51,52,53].

At the same time, a strong relationship has been observed between the decrease in academic performance and the school transition period according to the literature, although there is still no causal relationship between them [54]. The drop in academic performance may also be due to declining self-concept, motivation, social and emotional difficulties, and the increasingly competitive environment when faced with higher courses [55].

The results show a statistically significant relationship between anxiety and negative emotions with respect to academic performance, showing a negative correlation, as in the study by Jordan, JA et al. [56] where it mentions a negative correlation between stress and academic performance; which can be explained by the hypothesis of cognitive damping [57].

### Limitations

The statistically significant differences between gender and the academic year, in terms of emotions related to exams, highlight the importance of carrying out an evaluation that highlights the emotional difficulties that each student suffers when undergoing an exam, in order to intervene effectively and that this does not negatively affect their academic performance [4,7]. Therefore, the EES facilitates a correct evaluation of the emotions related to the exams in the context of primary and secondary education. But although the EES presents sufficient reliability and validity, it also presents a series of limitations regarding the difficulty of generalizing the results to other population groups, which compromises the external validity (population and ecological) of the questionnaire, as well as the difficulty of establishing convergent and discriminant validity evidence. Therefore, the replication and expansion of the study would allow the possibility of establishing new evidence of discriminant and convergent validity.

## 5. Conclusions

The present study demonstrates the psychometric properties of EES scale to assess the exam-related emotions of primary and secondary school students. Therefore, this research provides an instrument with scientific rigor and with adequate reliability and validity to be able to know the emotions that primary and secondary school students suffer when they are subjected to evaluation processes in the academic context. Likewise, the statistically significant differences found in the research, in terms of emotions and academic performance of students, as well as in gender and academic year, provide current information, relevant for the application of interventions aimed at preventing negative emotions and anxiety related to exams from negatively impacting on academic performance and leading to school failure.

However, future researchers should analyze the EES factor structure with different student populations, with the aim of improving the tool. In addition, future studies should analyze the structure of invariance with respect to other variables like region, County, center size or type of center (e.g., private, public or concerted).

## Figures and Tables

**Table 1 ijerph-19-06770-t001:** Exploratory factor analysis of the “*Exams-related Emotions Scale (EES)*”.

	Component			Communalities
	1	2	3	
After the exam I feel embarrassed.	0.795			0.653
During the exam I feel embarrassed.	0.795			0.635
Before the exam I would prefer not to take the exam because I have lost all hope.	0.766			0.642
During the exam I begin to think that no matter how hard I try, I will never pass the exam.	0.760			0.643
I’m embarrassed by my notes.	0.748			0.565
During the exam I feel so resigned that I have no energy.	0.743			0.608
During the exam I am embarrassed how badly I prepared.	0.742			0.557
During the exam, I feel like dropping out.	0.734			0.609
Before the exam I get depressed because I feel like I don’t have much hope of passing the exam.	0.711			0.636
When I get a bad grade I wish I didn’t have to look my teacher in the face again.	0.665			0.455
During the exam I get angry.	0.573			0.341
During the exam it seems to me that the questions are unfair.	0.457			0.211
During the exam I think I can be proud of my knowledge.	13	0.804		0.648
After the exam I am very satisfied with myself.	14	0.762		0.601
During the exam I am very confident.	15	0.760		0.611
During the exam I am happy to be able to take the exam.	16	0.760		0.579
During the exam, as I hope to pass I am motivated to try hard.		0.742		0.558
After the exam I am brimming with enthusiasm.	18	0.738		0.562
Before the exam, as I enjoy preparing for exams I am motivated to do more than necessary.	19	0.695		0.493
Before the exam I am so proud of how I prepared that I want to start the exam right now.	20	0.692		0.486
Before the exam I think optimistically about the exam.	21	0.649		0.439
After the exam I feel very relieved.	22	0.612		0.441
After the exam I can finally breathe easy again.	23	0.491		0.417
After the examination the nerves in the stomach disappear.	24	0.463		0.267
As I begin the exam my heart begins to accelerate.	25		−0.776	0.610
During the exam I am very nervous.	26		−0.796	0.679
Before the exam I feel nervous and restless.	27		−0.763	0.600
During the exam my hands tremble.	28		−0.708	0.548
Before the exam I am worried if I will have studied enough.	29		−0.660	0.444
During the exam I am worried about whether I will pass the exam.	30		−0.637	0.453
Before the exam I get so nervous that I wish I could miss the exam.	31		−0.623	0.545
Extraction Method: Principal Component Analysis Rotation method: Oblimin with Kaiser normalization				

**Table 2 ijerph-19-06770-t002:** Goodness of fit indices of models of emotion in exams proposed.

Models		χ^2^	CMIN/DF	CFI	TLI	RMSEA	SRMR
M1	3 related factors	*p* < 0.001	1.911	0.913	0.904	0.058	0.063
M2	2 related factors 1 independent (Anxiety)	*p* < 0.001	2.304	0.875	0.863	0.069	0.177
M3	2 related factors 1 independent (positive emotions)	*p* < 0.001	2.120	0.900	0.901	0.069	0.169
M4	3 independent factors	*p* < 0.001	2.438	0.863	0.849	0.073	0.226

Note: CMIN = chi-square ratio over degrees of freedom; CFI = comparative adjustment index; TLI = Tucker-Lewis index; RMSEA = mean square error of approximation; SRMR = standardized residual mean square root.

**Table 3 ijerph-19-06770-t003:** Factorial loads of three Related Factors Model of the Exams-related Emotions Scale (EES).

F1 Negative Emotions			F2 Positive Emotions			F3 Anxiety		
Item	Emotion	Estimate	Item	Emotion	Estimate	Item	Item	Estimate
**1**	Shame4	0.693	**13**	Pride2	0.814	**25**	Anxiety6	0.691
**2**	Shame2	0.734	**14**	Pride3	0.702	**26**	Anxiety4	0.815
**3**	Hopelessness2	0.813	**15**	Hope2	0.729	**27**	Anxiety4	0.716
**4**	Hopelessness3	0.812	**16**	Enjoy2	0.721	**28**	Anxiety7	0.682
**5**	Shame5	0.717	**17**	Hope3	0.752	**29**	Anxiety2	0.596
**6**	Hopelessness5	0.752	**18**	Enjoy3	0.652	**30**	Anxiety5	0.573
**7**	Shame3	0.662	**19**	Enjoy1	0.595	**31**	Anxiety3	0.732
**8**	Hopelessness4	0.727	**20**	Pride1	0.603			
**9**	Hopelessness1	0.792	**21**	Hope1	0.616			
**10**	Shame6	0.639	**22**	Relief1	0.534			
**11**	Anger1	0.593	**23**	Relief3	0.487			
**12**	Enfado2	0.548	**24**	Relief2	0.521			

**Table 4 ijerph-19-06770-t004:** Multigroup analysis of invariance by gender.

Models	χ^2^	g.l	χ^2^/g.l	Δχ^2^	*p*	Δgl	CFI	TLI	SRMR	RMSEA
Model 1	1540.126	844	1.825	-	-	-	0.890	0.880	0.088	0.046
Model 2	1547.013	872	1.774	6.887	1.000	28	0.894	0.886	0.089	0.044
Model 3	1549.138	878	1.764	9.012	1.000	34	0.894	0.888	0.094	0.044
Model 4	1569.753	918	1.710	29.626	1.000	74	0.897	0.896	0.097	0.042

Model 1 = No restrictions. Model 2 = Measurement weights. Model 3 = Structural covariances. Model 4 = Measurement residues.

**Table 5 ijerph-19-06770-t005:** Independent samples *t*-Test between EES and gender results.

EES	Gender				Independent Samples *t*-Test	
	Male (*n* = 284)		Female (*n* = 278)			
	M	SD	M	SD	*t*	*p* Value (Bilateral)
**Negative Emotions**	23.69	10.19	25.16	11.01	−1.636	0.102
**Positive Emotions**	42.81	9.35	40.65	9.76	2.680	0.008
**Anxiety**	20.74	7.21	23.99	6.80	−5.502	0.000

Note: EEs = Exams-related Emotions Scale.

**Table 6 ijerph-19-06770-t006:** Results of the ANOVA test between the EES and the academic year.

	Course	N	Mean	Standard Deviation	F	Sig. (Bilateral)
**Negative Emotions**	**5º EPO**	148	23.46	10.62	9.201	0.000
	**6º EPO**	186	21.88	10.14		
	**1º ESO**	134	27.14	10.82		
	**2º ESO**	94	27.06	9.90		
**Positive Emotions**	**5º EPO**	148	43.30	9.63	16.056	0.000
	**6º EPO**	186	44.41	8.90		
	**1º ESO**	134	39.14	9.14		
	**2º ESO**	94	37.71	9.29		
**Anxiety**	**5º EPO**	148	22.45	7.13	4.721	0.002
	**6º EPO**	186	20.87	7.32		
	**1º ESO**	134	23.67	7.04		
	**2º ESO**	94	23.24	6.80		

**Table 7 ijerph-19-06770-t007:** Pearson Test Results.

	Academic Performance (Average Grade)	*p*
**Negative Emotions**	−0.406	<0.001
**Positive Emotions**	0.385	<0.001
**Anxiety**	−0.212	<0.001

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
