# Peer review of "Psychometric Analysis and Contribution to the Evaluation of the Exams-Related Emotions Scale in Primary and Secondary School Students"

_ijerph, 2022, doi:10.3390/ijerph19116770_

Round 1

Reviewer 1 Report

The manuscript entitled "Validation of the Exams-related Emotions Scale in primary and secondary school students" aimed to validate the Exams-related Emotions Scale.

The study's theme is very interesting, and the manuscript is generally well written. Nonetheless, there are such limitations that I have doubts as to whether this manuscript can be considered an effective validation of the tool. 

The manuscript does not mention the use of a scale to calculate divergent validity, but only that of content. Lacking this aspect, I don't think it can be considered a "validation". 

For this reason, I believe that the title and the whole purpose of the research should change into a "contribution" to the evaluation of the EES.

I therefore believe that the manuscript should be revised in its entirety and substantially revised, starting from the title and the intentions of the authors.

Furthermore, the following small doubts must be resolved.

Abstract

Pag. 1, line 22: In the abstract, the average and the Standard Deviation of the age and the percentage of males and/or females in the sample are not reported.

Method

Pag. 3, line 102: The participant section must include information about the sample (including Standard Deviation which is missing here). In fact, what does DT stand for? If it is not clear, better specify, and if it was a typo, correct it. It is also correct to report the percentage of each subgroup that makes up the sample. Finally, in English, the sentence after the period cannot begin with a number written in figures, but in words.

Reviewer 2 Report

Thank you for the opportunity to review the paper entitled “Validation of the Exams-related Emotions Scale in primary and secondary school students.” In this validation study, the authors conduct an EFA and CFA to examine the factor structure of the Exams-related Emotions Scale (EES). An independent samples t-test was used to examine the difference in EES scores between males and females, and an ANOVA was used to examine the difference in EES scores between students from different grade levels. Results support a three-factor structure; males scored higher on positive emotion and lower anxiety; negative emotions appeared to be related to the increase of exams in secondary courses, but positive emotions were related to higher scores in primary courses. The authors did a nice job of describing the method used. While there are some small typos in the paper, it is generally well-written and appears to need only some light copyediting work. I have only some minor feedback:

Was there any missing data? If so, please describe how it was handled in the method section.

  • It would be helpful to include a justification for the choice to use an oblique (i.e., oblimin) rotation method for your EFA.
  • I appreciate the authors’ work on this study and wish them all the best in their continued contributions to this area.

Round 2

Reviewer 1 Report

I noted with regret that only a few minor notes were accepted by the authors in the review process. The biggest limitation I find is that there are no convergent and discriminant measures of validity in this study. This aspect is mentioned in the discussion section, but it has no repercussions on the nature of the paper. Lacking this aspect, the study can't be considered a "validation". For this reason, I advised changing the title and the aims of the research into a "contribution" to the evaluation of the EES. Furthermore, this limitation should be further emphasized by creating a separate section for "limitations". I, therefore, believe that the manuscript should be revised and improved again, also concerning the title and the aims of the authors.
